# OFASYS: A MULTI-MODAL MULTI-TASK LEARNING SYSTEM FOR BUILDING GENERALIST MODELS

## ABSTRACT

Generalist models, which are capable of performing diverse multi-modal tasks in a task-agnostic way within a single model, have been explored recently. Being, hopefully, an alternative to approaching general-purpose AI, existing generalist models are still at an early stage, where modality and task coverage is limited. To empower multi-modal task-scaling and speed up this line of research, we release a generalist model learning system, OFASys, built on top of a declarative task interface named *multi-modal instruction*. At the core of OFASys is the idea of decoupling multi-modal task representations from the underlying model implementations. In OFASys, a task involving multiple modalities can be defined declaratively even with just a single line of code. The system automatically generates task plans from such instructions for training and inference. It also facilitates multi-task training for diverse multi-modal workloads. As a starting point, we provide presets of 7 different modalities and 23 highly-diverse example tasks in OFASys, with which we also develop a first-in-kind, *single* model, OFA+, that can handle text, image, speech, video, and motion data. The single OFA+ model achieves 95% performance in average with only 16% parameters of 15 task-finetuned models, showcasing the performance reliability of multi-modal task-scaling provided by OFASys.

## 1 INTRODUCTION

Deep learning researches have achieved great success in designing curated model structures, data representations, and training objectives to pursue ultimate performance for a single model on a narrow field of tasks Srivastava et al. (2022). Recently, generalist models, e.g., OFA Wang et al. (2022a), Flamingo Alayrac et al. (2022a), GATO Reed et al. (2022a) and Unified-IO Lu et al. (2022), have been working towards a different vision of performing diverse multi-modal tasks in a task-agnostic way within a single model.

Inspired by the success of large language models Brown et al. (2020); Raffel et al. (2020); Chowdhery et al. (2022), a generalist model expresses any task intention via natural language and provides unified representations for the same modality data across all tasks. Such general task representation and task-agnostic learning, which is considered the prerequisites to approaching general-purpose AI Minsky (1988), have inspired the potential of generalizing to unseen tasks even with different modality compositions. Despite the disadvantage that no task-specific model structures are introduced in the finetuning stage, these generalist models have demonstrated unprecedentedly the possibilities of achieving superior finetuning performance, by task-agnostic multi-modal and multi-task pretraining.

However, being at an early stage, the research on generalist models is much more complicated in engineering compared to that on task-specific models, since it requires systematic management of the relationships among multiple modalities and tasks, in addition to the neural architectures and the single task optimization. This engineering complexity becomes intractable as the modality and task population grow to larger scale. While both open-source tensor libraries, such as TensorFlow Abadi et al. (2016) and PyTorch Paszke et al. (2019), and domain-specific libraries, such as Transformers Wolf et al. (2020) and MMDetection Chen et al. (2019), greatly expedite the development of task-specific models and applications, there

Table 1: Approaches to building generalist models. `OFASys` covers all of the approaches. CLM stands for causal language modeling, S2S stands for sequence-to-sequence learning, and DDPM stands for denoising diffusion probabilistic modeling. The numbers in parentheses are the numbers of supported modalities. The unique modalities supported by `OFASys` are `AUDIO` and `MOTION`.

| Model | Paradigm | Supervised | Multi-Task | Multi-Modal |
|---|---|:---:|:---:|:---:|
| GPT | CLM | ✗ | ✗ | ✗ |
| PaLM | CLM | ✗ | ✗ | ✗ |
| FLAN | CLM | ✓ | ✓ | ✗ |
| Flamingo | CLM | ✗ | ✓ | ✓(3) |
| Gato | CLM | ✓ | ✓ | ✓(4) |
| T0 | S2S | ✓ | ✓ | ✗ |
| OFA | S2S | ✓ | ✓ | ✓(3) |
| Unified-IO | S2S | ✓ | ✓ | ✓(4) |
| `OFASys` | CLM, S2S, DDPM | ✓ | ✓ | ✓(7) |

is currently no designated system that provides neat abstractions and tools for task-agnostic generalist model learning.

The inherent diversity and heterogeneity of multi-modal multi-task learning stands out as a major obstacle in establishing such a system. In conventional practice, different tasks may require a different model structure and a different training pipeline, all contributing to the state-of-the-arts performance per task. Although the particular traits of the specific task can be efficiently addressed, it proves hard to scale, as each new task would demand a new system design.

Recently, studies have shed light on that issue and provided an alternative, scalable way to multi-modal multi-task learning, drawing inspirations from (a) the task generalization capabilities demonstrated by pretrained language models Brown et al. (2020); Sanh et al. (2022) and (b) the success of the Transformer architecture in universal multi-modal learning Wang et al. (2022a;b); Radford et al. (2021); Chen et al. (2022b). This motivates us to decouple the task representation from its model implementation, which enables researchers to investigate multi-modal task scaling and the underlying model compositions independently.

In light of this, we propose `OFASys`, a system designed for building generalist models via multi-modal multi-task learning. The goal of `OFASys` is to facilitate the research of multi-modal multi-task learning with a concise, flexible user interface and a modularized, reusable system design.

For users, `OFASys` enables both fast prototyping and in-depth customization via a declarative interface called "*Multi-Modal Instruction*". The instruction describes a task using natural language with multi-modal data placeholders called "*slots*". Users can declare a new task in just a single line of code, or customize task-specific processing and new modalities, which is seamlessly combined with the instruction interface. As a starting point, `OFASys` comes with 7 modality presets, *i.e.*, `TEXT`, `IMAGE`, `AUDIO`, `VIDEO`, `STRUCT`, and `MOTION`, which compose 23 example tasks that vary widely in modality compositions and task objectives. A brief demonstration of the available modalities and representative example tasks is shown in Figure 1.

To realize the goal of `OFASys`, the system design, which disentangles the complexity of implementing task-specific pipelines, forms reusable component hierarchies in different granularity: (a) for different tasks, the model and the training/inference components can form training/inference pipelines, (b) for a single task, the universal model and the modality-specific model components can form a multi-modal model computing pipeline, and (c) for a slot in a task, the pre-/post-processors and the adapters can form modality-specific data pipelines. In addition, the multi-task engine provides support to efficient multi-task training.

`OFASys` enables us to train a series of specialist and generalist models, which we name OFA+. The OFA+ (Specialist) models showcase the capability of the instruction interface, with

**Instruction (TEXT Only)**

```
what is the summary of article "[
↪ TEXT:src]"? -> [TEXT:tgt]
```

**Data**

*poland 's main opposition party tuesday endorsed president lech walesa in an upcoming presidential run-off election after a reformed communist won the first round of voting .*

**Result**

polish opposition endorses walesa in presidential run-off

(a) Text Summarization

**Instruction (AUDIO & TEXT)**

```
[AUDIO:wav] what is the text
↪ corresponding to the voice? -> [
↪ TEXT:text,preprocess=text_phone]
```

**Data (Illustration)**

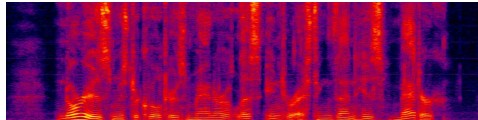

**Result**

nor is mister klohs manner less interesting than his manner

(b) Automatic Speech Recognition

**Instruction (IMAGE, TEXT & BOX)**

```
[IMAGE:img] which region does the
↪ text "[TEXT:cap]" describe? -> [
↪ BOX:box]
```

**Data & Result**
cap: *taxi*

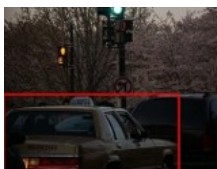

(c) Image Grounding

**Instruction (MOTION & TEXT)**

```
motion capture: [TEXT:text] -> [
↪ MOTION:bvh_frames]
```

**Data**
*run and stop*
**Result (Illustration)**

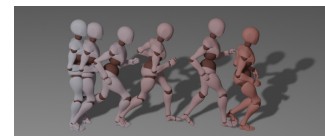

(d) Text-to-Motion Synthesis

Figure 1: `OFASys` enables extremely diverse compositions of modalities for tasks via the instruction interface. Results are generated by models trained by `OFASys`.

which one can expediently compose new tasks. The OFA+ (Generalist) model spans over 7 modalities and is trained on 17 tasks, suggesting that training a single model with more modalities and tasks is achievable. Moreover, the OFA+ (Generalist MoE) models with a sparsely-activated universal model demonstrates promising results. By using a simple multi-task schedule, OFA+ (Generalist MoE) achieves 95 % of the performance with just 16 % of the parameters of task-finetuned models.

The contribution of `OFASys` is summarized as follows:

- `OFASys` is an open-source generalist model learning system that is designed for multi-modal task-scaling. `OFASys` offers to the community 7 pre-defined modalities and 23 example tasks for reuse in multi-modal multi-task learning research.

- `OFASys` provides an easy-to-use declarative interface that decouples task definitions from task implementations, thus enabling fast prototyping. The design of `OFASys` disentangles the complexity of single task pipelines into a hierarchy of components, which can be easily reused, customized, and replaced for in-depth research.

- `OFASys` trains the first-in-kind models named OFA+ that can understand and generate text, image, speech, video, and motion data. The checkpoints are publicly available upon request.

## 2   Usage with Declarative Multi-Modal Instruction

In the following, we briefly show the basic high-level usage of `OFASys` about how to represent, train, and conduct inference on multi-modal tasks with the *multi-modal instruction* interface. For more usage illustrations, please refer to Appendices B.1 to B.3.

A multi-modal instruction is a descriptive line of code that specifies what the task is supposed to do and what kinds of modality of data are involved. With the help of instructions, we can create multi-modal tasks as follows, for which `OFASys` determines the model structure and training/inference–related components automatically according to the preset choices or the given customized implementations:

```python
from ofasys import Task, GeneralistModel, Trainer
task1 = Task(instruction1)
task2 = Task(instruction2)
instruction = "[IMAGE:img] what does the image describe? -> [TEXT:cap]"
```

The preceding tasks can then be bound to some dataset and join the training of a generalist model as

```python
task1 = task1.add_dataset(dataset1)
task2 = task2.add_dataset(dataset2)
model = GeneralistModel()
Trainer().fit(model=model, tasks=[task1, task2])
```

The "*GeneralistModel*" has various kinds of implementations, which are discussed in Sec. 4.

Such an instruction interface also enables zero-shot inference with generalist model checkpoints, which allows one to evaluate its generalization ability on unseen tasks:

```python
instruction = '[IMAGE:img] what does the image describe? -> [TEXT:cap]'
data = {'img': 'image_1.jpg'}
output = GeneralistModel.from_pretrained('multitask.pt').inference(instruction, data=data)
```

The term "*declarative*" indicates that a task is created by its formulation rather than the control flow. Researchers can fix the provided model set while only study task-scaling problems, or fix the task set to study the underlying model structure. Different from task-specific libraries Wolf et al. (2020); Chen et al. (2019), `OFASys` emphasizes on using the same universal model to handle both pretraining and finetuning tasks.

## 3   User Interface

Now we introduce the details of the instruction user interface, including its formulation and several instruction examples that illustrate the purpose of each consideration.

### 3.1   Core Concepts

An **instruction** contains multiple segments, each of which is either "*plain texts*" describing the task goals or a "*meta_slot*" describing the multi-modal data. A "*meta_slot*" can be either a "*slot*" or an "*expanded_pattern*" consisting of slots.

A **slot**, identified by square brackets from plain text segments, is the basic processing unit of `OFASys`. A slot consists of a "*type*", a "*name*", and optional "*attributes*" of "*key=value*" pairs. `OFASys` uses these metadata to configure the modality-aware data processing. The type specifies the modality of the data. The name is used to retrieve data from the data source. The attributes customize data processing.

The **expanded pattern** allows the system to register more syntax on the instruction. There are some typical patterns in `OFASys`. For example, an arrow "->" is for the system to identify slots on the encoder from decoder, and it can be omitted for decoder-only models.

## 3.2 Instruction Examples

In the following, we demonstrate several representative tasks written by instructions, to help understand how these concepts work in practice.

**Slot Type and Name.** A slot is only attached with one unique modality, whose type is represented as the slot type. The slot name is used mainly for field mapping in dataset.

> *Illustration 1.* Basic Image Captioning
> ```
> [IMAGE:img] what does the image describe? -> [TEXT:cap]
> ```
>
> The two sentences separated by "->" describe the task input and its desired output, respectively. In this case, "[IMAGE:img]" specifies that there is an image input bound to a data column named `img` in the dataset. The plain texts in the instruction indicate the task is about captioning an image. The output of the task is a text sequence, which is the `cap` column in the dataset.

**Attributes.** Attributes allow fine-grained control over a certain slot. Users can exploit built-in attributes or implement customized ones.

> *Illustration 2.* MNLI with Prompt Prefix
> ```
> can text1 [TEXT:s1] imply text2 [TEXT:s2]? -> can text1 [TEXT:s1,no_loss] imply
> ↪ text2 [TEXT:s2,no_loss] ? [TEXT:label,closed_set]
> ```
>
> For text classification tasks, *e.g.*, MNLI Williams et al. (2018), we find it helpful to repeat the source as prompt prefix in the output Wang et al. (2022a). However, the decoder slots default to the cross-entropy loss, which can be disabled using shortcut `no_loss`. The prefixes are also ensured to be generated by `OFASys` in inference. MNLI also has a limited label space, which can be constrained using the attribute `closed_set`. Enumeration of the closed set is specified in the task configuration.

Overall, the instruction formulation (together with appropriate implementation) allows the expression of diverse task paradigms. More details are shown in Appedix A In theory, it can not only describe tasks of CLM, S2S, and DDPM paradigms shown in Tab. 1, but also support paradigms such as soft prompt tuning in NLP Li & Liang (2021) through custom slots, CLIP-style contrastive learning Radford et al. (2021) through custom criteria, and Flamingo-style in-context learning Alayrac et al. (2022a) with custom instruction parsing.

## 4 System Design and Implementation

In this section, we start from the motivation of the system design with its relation to the instruction interface, and further introduce the details of the system implementation to reveal the considerations in realizing the design.

### 4.1 Scaling Challenges in Training Multi-Modal Multi-Task Models

Frameworks such as fairseq Ott et al. (2019) and transformers Wolf et al. (2020), have *de facto* standardized and streamlined the procedures of a specialized group of deep learning methods, reducing the development cost of handling task-specific training and inference. However, these frameworks are not sufficient with the surge of multi-modal and multi-task learning Wei et al. (2021); Sanh et al. (2022); Wang et al. (2022c;a); Reed et al. (2022b); Alayrac et al. (2022b); Lu et al. (2022), which faces profound challenges in terms of the heterogeneity of multi-modal data, the diversity of task formulation, and the complexity of scalable schedule of computation. Researchers have to, on their own, (a) implement specific data processing procedures for every task, (b) adapt the model structure and computation to each task with different feature extractors and losses, (c) manually determine the task precedence during optimization for the model performance, and (d) manage sample batching,

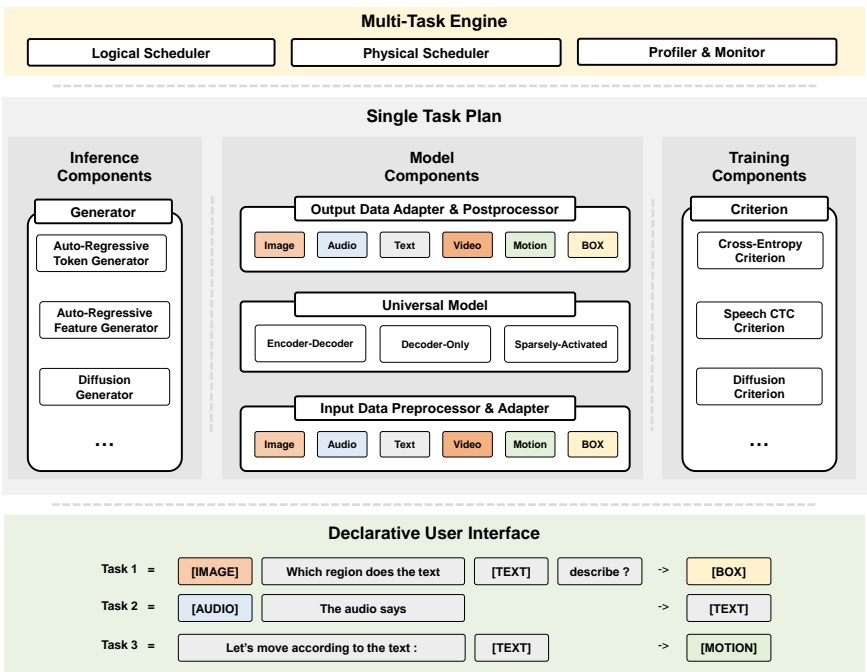

Figure 2: Overview of main components and their organization in OFASys. The declarative user interface is realized with a hierarchy of reusable components. Based on instructions, OFASys automatically generates plans for single task execution in both training and inference. Moreover, the multi-task engine provides support to efficient multi-task learning.

op-level replacement of highly diverse task workloads in distributed environment, for training efficiency.

To facilitate multi-modal multi-task learning, we develop the One-For-All System (OFASys) to seamlessly integrate multiple modalities and multiple tasks into a single framework with universal models, as well as automatic task schedulers to manage multi-task execution.

## 4.2 SYSTEM OVERVIEW

We illustrate the overview of the system with an example of the data flow of a single task to show how heterogeneous inputs are processed by OFASys, and further illustrate the design on managing multiple tasks w.r.t. to single task plans. Detailed component introduction is provided in Appendix C.

**Single Task Plan.** OFASys accesses the task definition and task data through instructions, as introduced in Sec. 3. To realize the instruction as viable computation, OFASys parses the instructions into task plans[1]. In each plan, there is a model hierarchy, consisting of modality-specific preprocessors/postprocessors and adapters, as well as a modality-agnostic computation model. For an example, the system first retrieves data specified by the slots and dispatches them to corresponding preprocessors to convert them into common ML input types, e.g., tokens for texts and fbank features for audio. Then, the preprocessed data are dispatched to the corresponding adapters for feature extraction, and the output features are joined to form sequences of representations for the universal model. These steps ensure a unified data format for the universal model but with reusable and composable data processing pipelines. The universal model is namely a general module for fusing multimodal inputs and generating outputs. As the inputs and the outputs are consistently being representation sequences, the implementation of the universal model is highly versatile, regardless of the

---

[1]We borrow the term *plan* from the *logical/physical plan* in database literature Elmasri & Navathe (2000).

modality intricacies. The outputs of the universal model is finally postprocessed by the adapters and postprocessors, in order to generate content consistent with the input formats. Stage-wise components, including criteria and generators, provide support in training and inference, which have a variety of out-of-the-box implementations. In this way, different multi-modal data can go through the system with consistent inner interfaces to improve development efficiency.

**Multi-Task Plan.** In multitask learning, there are multiple such plans parsed from the instructions. There are two problems to be dealt with: (a) how a single model is used with multiple plans and (b) how multiple tasks are optimized and executed together. For the first problem, `OFASys` shares the trainable parameters of the adapters and the universal model by default, such that each parameter can be optimized on as many examples as possible. To be specific, the modality-specific components are shared across the same modality and the universal model is shared across all tasks. For the second problem, the task scheduler is in charge of the management of task precedence and the optimization of the execution details of multi-task workloads with two levels of abstraction. `OFASys` automatically combines all the plans to form a multi-task plan with a *logical* scheduler and then the workflow is arranged on physical devices with a *physical* scheduler. More details are given in Appendix C.4.

## 5 APPLICATION EXAMPLE: THE OFA+ MODELS

We train the generalist models, referred to as OFA+ (Generalist) and OFA+ (Generalist+MoE), which can handle text, image, speech, video and motion data all-in-one for the first time, using `OFASys`.

### 5.1 SETTINGS

Apart from models finetuned on each of the task, OFA+ (Specialist), we train two versions of generalist models as the validation of the system design and implementation which we call OFA+ (Generalist) and OFA+ (Generalist MoE).

The first model, OFA+ (Generalist), is of similar structure to OFA-base Wang et al. (2022a) and is initialized from the pretrained checkpoints of OFA-base. It has 270 M parameters in total, of which 90 M are modality-specific parameters. The second model, OFA+ (Generalist MoE), is also based on OFA-base. Especially, the model structure is augmented with a sparsely-activated implementation of the universal model. An existing similar implementation is VLMO Wang et al. (2021), which distributes FFN in transformer layers based on image and text at the bottom layer and visual language at the top layer. Different from VLMO, we distribute FFN based on different modalities of the slot, such as text, image, speech, video, and motion, at each layer in the encoder. There are 455 M parameters in total, of which 275 M are modality-specific.

Both models are trained with 17 downstream tasks together involving data from 7 modalities. For a detailed list of the task mixture, please refer to Appendix F. In evaluation, both models do not go through task-specific finetuning. We report results on the validation sets, except for VQA v2, where test-dev is commonly used as validation, and 2 tasks that do not come with validation set, *i.e.*, grounded image captioning and text-to-motion synthesis.

For OFA+ (Generalist) and OFA+ (Generalist MoE), we use the AdamW Ilya & Frank (2019) optimizer with $(\beta_1, \beta_2) = (0.9, 0.999)$ and $\epsilon = 1 \times 10^{-8}$ to train the model. We set the peak learning rate to $3 \times 10^{-4}$, and apply a scheduler with linear decay with a warmup ratio of 0.01 to stabilize the learning. For more detailed settings, such as batch size for each task, please refer to Tab. 4 in the appendix. Both models are trained using 32 NVIDIA A100 80GB GPUs.

### 5.2 RESULTS AND FINDINGS

We report results on 15 tasks used in our training setup to evaluate a single jointly-trained model. In fact, to our knowledge, it is the first time that data from those 7 modalities are used together to train a single model, thanks to the support from `OFASys`.

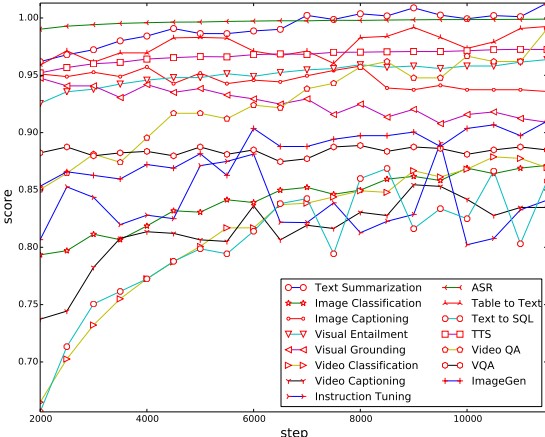

Figure 3: Learning curves in terms of metrics for OFA+ (Generalist). The y-axis represents the percentage of the performance w.r.t. the corresponding specialist. The results of TTS and ASR are linearly transformed so that for all results, higher is better. As we can see, different tasks have divergent learning speed and reach maximum performance at different steps.

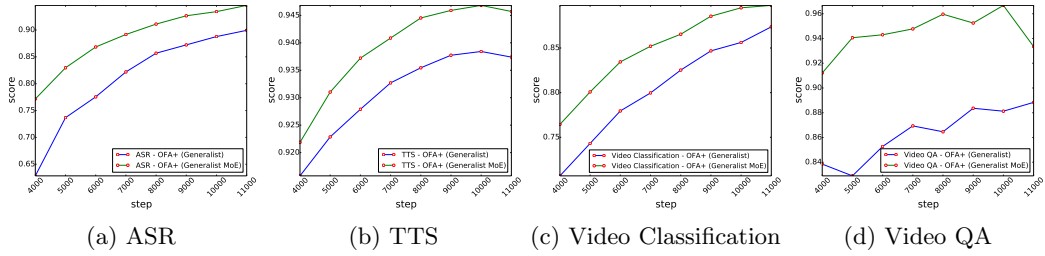

(a) ASR                (b) TTS                (c) Video Classification        (d) Video QA

Figure 4: Learning curves in terms of metrics for OFA+ (Generalist) and OFA+ (Generalist MoE) on representative tasks. Arranged the same with Figure 3. It can be seen that the MoE version converges faster and better in modalities such as audio and video.

In Figure 3, we show the learning curves of different tasks in the OFA+ (Generalist) model. We can see that different tasks have divergent learning speed and may reach maximum performance at different steps, given a vanilla implementation of the universal model. In Figure 4, we further compare the learning curves of generalist models on different universal model structures. As we can see, OFA+ (Generalist MoE), which is slot-wise sparsely-activated, converges faster and better than the other, regarding to the video and the audio tasks. OFASys facilities such investigation of model structures by only modifying the universal model implementation or the model composition, while keeping the entire instruction set, *i.e.*, tasks, fixed.

We compare the validation results between specialist models and generalist models in Tab. 2. The OFA+ models trained on single tasks is referred to as OFA+ (Specialist). Note that due to the differences in model sizes, the experimental results are not precisely comparable. Comparing OFA+ (Generalist) and OFA+ (Generalist MoE), we can see that the latter performs much better on vision-related tasks. The overall average score also understandably favors OFA+ (Generalist MoE). With only 16 % parameters of the 15 task-finetuned specialist models, OFA+ (Generalist MoE) still achieves 95 % of the specialists' performance in average, showcasing the potential of multi-modal task-scaling with OFASys.

As the OFA+ (Generalist) and OFA+ (Generalist MoE) models have scaled the number of modalities to a new level, there is currently no precisely-comparable models from other studies. The most comparable method is Unified-IO Lu et al. (2022) which conducts text and image tasks in a similar finetuning-free manner to us. Compared to Unified-IO, generalist

Table 2: Experimental results on OFA+ (Specialist), OFA+ (Generalist), and OFA+ (Generalist MoE) according to the validation set (test-dev for VQA v2), with the exception of grounded image captioning and text-to-motion synthesis, which lack validation sets. ↑ indicates the higher the better; ↓ indicate the lower the better. R-L stands for ROUGE-LCS Lin (2004), Acc@0.5 stands for accuracy where IoU ≥ 0.5 is considered correct Yu et al. (2016), and EM stands for accuracy where an exact match is considered correct Zhong et al. (2020a). Results of Unified-IO are taken directly from Lu et al. (2022).

| Tasks | Datasets | Metrics | OFA+ (Specialist) 186 M × 15 | OFA+ (Generalist) 270 M | OFA+ (Generalist MoE) 455 M | UnifiedIO 776 M |
|---|---|---|---|---|---|---|
| *Text-only tasks* | | | | | | |
| Instruction Tuning | NatInst v2 | R-L ↑ | 30.5 | 27.0 | 27.7 | - |
| Summarization | Gigaword | R-L ↑ | 34.2 | 34.7 | 34.0 | - |
| *Image-related tasks* | | | | | | |
| Classification | ILSVRC | Acc ↑ | 83.3 | 72.6 | 79.0 | - |
| Visual Entailment | SNLI-VE | Acc ↑ | 88.9 | 85.8 | 86.2 | 86.1 |
| Captioning | COCO | CIDEr ↑ | 134.8 | 122.6 | 125.2 | 117.5 |
| Visual Grounding | RefCOCO | Acc@0.5 ↑ | 88.1 | 80.1 | 83.1 | - |
| VQA | VQA v2 | Acc ↑ | 77.3 | 68.2 | 71.7 | 67.8 |
| Image Generation | COCO | CLIPSIM ↑ | 0.317 | 0.289 | 0.294 | - |
| *Audio-related tasks* | | | | | | |
| ASR | LibriSpeech | WER ↓ | 7.5 | 8.5 | 8.1 | - |
| TTS | LJSpeech | $\mathcal{L}$ ↓ | 1.187 | 1.443 | 1.429 | - |
| *Video-related tasks* | | | | | | |
| Classification | Kinetics400 | Acc ↑ | 74.3 | 64.6 | 69.5 | - |
| Captioning | MSR-VTT | CIDEr ↑ | 70.8 | 59.1 | 63.0 | - |
| VQA | MSR-VTT QA | Acc ↑ | 42.1 | 41.7 | 40.0 | - |
| *Structure-related tasks* | | | | | | |
| Table-to-Text | DART | BLEU ↑ | 51.2 | 50.9 | 50.9 | - |
| Text-to-SQL | Spider | EM ↑ | 45.7 | 39.2 | 40.5 | - |
| | Average (Performance) | | 100 % | 91 % | 95 % | - |
| | Average (Model Size) | | 100 % | 10 % | 16 % | - |

OFA+ models are trained on a different mixture of tasks with more modalities and fewer vision-language tasks, also with substantially fewer parameters. The results demonstrate the generalist models trained by OFASys are better in terms of performance on the overlapping tasks.

In all, in application to OFA+, OFASys demonstrates evidently its functionality, scalability, and flexibility. The functionality is validated by training and conducting inference using a number of models on diverse tasks in single-task or multi-task settings. The full-fledged support allows meaningful exploration of generalist models beyond the language-vision or language-speech settings. The scalability lies in the composition of 23 diverse tasks over 7 modalities with a consistent declarative interface. They can theoretically be used to train a single model all together by reusing instructions even without writing new code. The flexibility is shown by the two versions of OFA+ (Generalist): the computation pipeline is decoupled reasonably enough for one to focus research on a part of the whole system. For example, one can focus on model structures w.r.t. modality by modifying only the universal model, without worrying breaking data processing pipelines.

## 6 Conclusion and Future Work

As generalist models attract increasing interests, the lack of designated system and library for multi-modal multi-task stands out as an obstacle in the path for rapid growth. OFASys is developed to match the need in multi-modal multi-task learning of extreme modality and task scaling. With OFASys, it is easy to (a) rapidly introduce new multi-modal tasks by defining a declarative instruction in a single line of code, and (b) easily introduce, reuse, and customize modality-specific components. The functionality is realized by a carefully-designed library structure that decouples the complexity of multi-modal multi-task learning into a hierarchy of components. We train a series of models named OFA+, using OFASys, and show

that it is achievable for a generalist model to understand data from more modalities and perform promisingly. In all, we hope OFASYs would push forward the research in multi-modal multi-task learning and facilitate the construction of generalist models that are even more general.

In this work, we focus on the importance of building a system that can support the development of generalist models, while we leave much room in improving the model performance and examining multi-task learning algorithms. For example, as related research on diffusion-based vision models progresses rapidly, it is interesting to see how to unify them appropriately with the system. The emergence of new learning paradigms, *e.g.*, multi-modal in-context learning through interleaved image-text data, also reveals opportunities in scaling the system. Our future work includes not only refining the system itself but also experimenting new algorithms on the system.

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

# A   INSTRUCTION EXAMPLES

In the following, we demonstrate several representative tasks written by instructions, to help understand how these concepts work in practice.

**Slot Type and Name.**   A slot is only attached with one unique modality, whose type is represented as the slot type. The slot name is used mainly for field mapping in dataset.

*Illustration 3.* Basic Image Captioning

```
[IMAGE:img] what does the image describe? -> [TEXT:cap]
```

The two sentences separated by "->" describe the task input and its desired output, respectively. In this case, "[IMAGE:img]" specifies that there is an image input bound to a data column named `img` in the dataset. The plain texts in the instruction indicate the task is about captioning an image. The output of the task is a text sequence, which is the `cap` column in the dataset.

**Attributes.**   Attributes allow fine-grained control over a certain slot. Users can exploit built-in attributes or implement customized ones.

*Illustration 4.* MNLI with Prompt Prefix

```
can text1 [TEXT:s1] imply text2 [TEXT:s2]? -> can text1 [TEXT:s1,no_loss] imply
↪ text2 [TEXT:s2,no_loss] ? [TEXT:label,closed_set]
```

For text classification tasks, *e.g.*, MNLI Williams et al. (2018), we find it helpful to repeat the source as prompt prefix in the output Wang et al. (2022a). However, the decoder slots default to the cross-entropy loss, which can be disabled using shortcut `no_loss`. The prefixes are also ensured to be generated by OFASys in inference. MNLI also has a limited label space, which can be constrained using the attribute `closed_set`. Enumeration of the closed set is specified in the task configuration.

**Variable-Length Slots.**   It is not uncommon for a task to generate several outputs of the same type, *e.g.*, object detection in computer vision. OFASys can support such type of tasks with variable-length slots using the interleaved_pattern.

*Illustration 5.* Object Detection with variable-length slots.

```
[IMAGE:img] detect the objects in the image. -> [[BOX][TEXT]]*
```

The output in object detection is normally a bounding box with its label. However, an image may contain several objects, whose number is not known in advance.

**Custom Slot.**   One can define their own type of slots to do interesting stuff. The example in Illus. 6 showcases the versatility of the slot concept. For guidance on how to support a new type of slot, please refer to Appendix B.3.

*Illustration 6.* Image Captioning with Prompt-Tuning.

```
[IMAGE:img] [PROMPT:pt,len=100,prefix-tuning] -> [TEXT:cap]
```

The instruction specifies a job with the customly-defined "*PROMPT*" slot type. OFASys registers "*len=100*" learnable embeddings to each layer using the name "*pt*", appends them to the source sequence for each example, and trains them using "*prefix-tuning*" Li & Liang (2021)

The preceding examples demonstrate the generality and versatility of the proposed user interface, which enables expedient task prototyping. Overall, the instruction formulation (together with appropriate implementation) allows the expression of diverse task paradigms.

In theory, it can not only describe tasks of CLM, S2S, and DDPM paradigms shown in Tab. 1, but also support paradigms such as soft prompt tuning in NLP Li & Liang (2021) through custom slots, CLIP-style contrastive learning Radford et al. (2021) through custom criteria, and Flamingo-style in-context learning Alayrac et al. (2022a) with custom instruction parsing.

## B  ADVANCED USAGE

In Sec. 2, we illustrate the high-level basic usage of OFASys. As we can see, a multi-modal task can be defined in a single line of code, which facilities task scaling via comprehensive default implementations. However, it is often one's desire to control the system with finer-granularity. OFASys provides two ways to achieve this goal, the YAML configuration that introduces structured parameters to instructions and a conventional imperative interface to programmatically define common procedures in computation pipelines. The first option is discussed in Appendices B.1 and B.2; and the second option is summarized in Appendix B.3.

### B.1  MULTI-TASK TRAINING WITH INSTRUCTIONS

Although it is straight-forward to use instructions directly in code, to achieve better reusability in training, OFASys can be also configured via configuration files. In experimentation, the configurations of different tasks, models, and trainers can be easily combined to build a new multi-modal multi-task learning system. OFASys organizes the training configuration files consistent with the routine in Sec. 2. The configuration files adopt the YAML format[2], which is human friendly and suitable for structured data.

### B.1.1  TASKS

For tasks, the configuration file should at least contain the instruction section and the dataset section. An example file for the captioning task is given in the following:

```yaml
# caption.yaml
instruction:
  - '[IMAGE:img] please use a short line to describe the image. -> [TEXT:cap]'
  - '[IMAGE:img] what does the image describe? -> [TEXT:cap]'

dataset:
  train_data: coco_2014/train
  valid_data: coco_2014/valid
  update_freq: 2
  micro_batch_size: 8

preprocess:
  image:
    patch_image_size: 480
  text:
    max_src_length: 128
    max_tgt_length: 20

criterion:
  label_smoothed_cross_entropy:
    label_smoothing: 0.1

evaluation:
  metrics:
    cider:
      target_field: cap
  generator_args: '{"beam":5,"max_len_b":16,"no_repeat_ngram_size":3}'
```

This configuration file presents a more structured view of the task pipeline components and showcases several advanced usage of OFASys, compared to Sec. 2, where all default values are used.

First, it can be seen that multiple instructions can be specified for a dataset. For each example in the dataset, OFASys currently randomly samples an instruction according to a

---

[2] https://github.com/yaml/yaml-spec

uniform distribution. However, it is also required that the instructions should be "*collation-compatible*", which means they should have the same number of slots and the sequence of slot types should be exactly the same.

Second, the dataset section specifies the paths of data used in training. As there are complications in declaratively describing how to construct the dataset, the data paths in configuration files should be either a HuggingFace Dataset spec, or a URI to a Tab-separated Values (tsv) file. The URI can be a local file path (optionally starting with file://) or a remote file path (where the protocol is a must). The tsv file should have a header and is parsed and processed as a `torch.utils.data.IterableDataset`. The dataset section also includes hyper-parameters in training the task, such as `micro_batch_size` and `update_freq`. The former is the per GPU batch size, and the latter is for gradient accumulation. Although they are related to optimization, they are included to streamline task composition in multi-task training. For more details, please refer to the documentation of `OFASys`.

Finally, there are optional configurations for task-specific pipelines. Slots of the same type share common preprocessor, postprocessor, and adapter configurations by default. These can be adjusted in the preprocess and the adapter sections. For example, the `max_src_length` sets the maximum length of every encoder `TEXT` slot to 128. The criterion used to obtain the loss can be automatically deducted from the instructions, which is the criterion based on cross entropy most of the time. A common usage is to add label smoothing normalization to the cross entropy, which is shown in the preceding example. The evaluation section tells how the performance of the task on this dataset can be evaluated. The `generator_args` is for the generator in inference to obtain the prediction and the metric section can include multiple metrics to evaluate the prediction against the ground truth in `target_field`.

For your information, the name in each section, *e.g.*, `image` and `text` in preprocess, `label_smoothed_cross_entropy` in criterion, and `cider` in metrics, is associated with the implementation via a hierarchical configuration registry. For a complete list of supported variants, please refer to the `OFASys` documentation.

The configuration file can be used to initialize a task with a dataset in one step:

```python
from ofasys import Task
task = Task.from_yaml("caption.yaml")
```

There are 23 readily-available task configuration files at the time of writing this manuscript. Please see Appendix E for reference.

### B.1.2 Universal Models

For universal models, the default configuration covers all aspects and no user input is required. The following file customizes the model implementation for all tasks:

```yaml
# model.yaml
model:
  arch: large
  use_fused: true
  freeze_encoder_embedding: true
  freeze_decoder_embedding: true
  encoder_drop_path_rate: 0.2
  decoder_drop_path_rate: 0.2
  layernorm_position: true
```

The `arch` specifies a set of configuration for model architecture, *e.g.*, embedding dimension and number of layers. The rest parameters further finetune the structure. For example, `use_fused` makes `OFASys` use fused CUDA kernel implementation for certain modules to improve computation performance on GPU devides. Encoder and decoder computation in training is adjusted via `freeze_*_embedding` and `*_drop_path_rate`, which indicates the adapters in the models are not optimized and a drop path normalization is used for both the encoder and the decoder. For complete configurable options, please also refer to the `OFASys` documentation.

The model can be initialized from files similarly to tasks:

```
from ofasys import GeneralistModel
model = GeneralistModel.from_yaml("model.yaml")
```

It should be well known that every tasks would share the same universal model structure.

### B.1.3   TRAINER

The trainer implements logistics in multi-task training. An example configuration file is given in the following:

```
# trainer_conf.yaml
common:
    fp16: true
    fp16_scale_window: 512
    log_interval: 10

distributed_training:
    find_unused_parameters: true

optimization:
    max_update: 10000
    clip_norm: 1.0
    lr: [1e-5]
    sentence_avg: false

optimizer:
    adam_betas: [0.9, 0.999]
    adam_eps: 1e-08
    weight_decay: 0.01

lr_scheduler:
    warmup_ratio: 0.06
```

The common section includes configuration for overall computation, profiler and monitor. The rest sections configure parameter optimization and others.

Considering memory efficiency, all tasks share a common optimizer and thus the same learning rate schedule, as most optimizers for multi-modal training estimate gradient momentums which diverge with different learning rates. Currently, the default optimizer is AdamW Ilya & Frank (2019) and the default learning rate scheduler is based on linear decay with warmup Liu et al. (2019). The options are being actively extended.

The trainer can be initialized as

```
from ofasys import Trainer
trainer = Trainer.from_yaml("trainer_conf.yaml")
```

OFASys also includes a launch script to conveniently combine the YAML configuration files in training. The launch script also wraps distributed training, which is highly environment-dependent. For related usage, please refer to the OFASys documentation.

### B.2   INFERENCE WITH INSTRUCTION

As tasks are expressed as instructions, OFASys supports task-agnostic inference using the trained models, which includes conducting novel multi-modal tasks.

As prerequisite, inference needs a pretrained model checkpoint. Especially the modality-related structures need training, if the modality is desired in inference.

```
from ofasys import OFASys
model = OFASys.from_pretrained('my-awesome-checkpoint.pt')
# Inference for image captioning
instruction = '[IMAGE:img] what does the image describe? -> [TEXT:cap]'
data = {'img': 'image_1.jpg'}
output = model.inference(instruction, data=data)
print(output.text) # the caption
```

As we can see, one only need to pass the instruction and the data for inference. The model checkpoints saved by `OFASys` keep all the training configurations so there is no need to include them again. Although the instruction is preferably similar to the ones used in training, there is no strict enforcement from the library side. The results can be retrieved with the slot type being the key.

`OFASys` also provides convenience methods to save `BOX` and `MOTION` output data. More inference examples are shown below:

Image Grounding (Auto-Regressive Token Generator):

```
instruction = '[IMAGE:img] which region does the text " [TEXT:cap] " describe? -> [BOX:region_coord]'
data = {'img': 'image_1.jpg', 'cap': 'hand'}
output = model.inference(template, data=data)
output.save_box('output_with_bbox.jpg')
```

Automatic Speech Recognition (Auto-Regressive Token Generator):

```
instruction = '[AUDIO:wav] what is the text corresponding to the voice? -> [TEXT:txt]'
data = {'wav': 'audio.flac'}
output = model.inference(template, data=data)
print(output.text)
# nor is mister quilters manner less interesting
# than his matter
```

Text-to-Motion Synthesis (Diffusion Generator):

```
instruction = 'motion capture: [TEXT:text] -> [MOTION:bvh_frames]'
data = {'text': 'run then jump'}
output = model.inference(template, data=data)
output.save_as_gif('run_jump.gif')
```

### B.3 Extensibility and Interoperability

Apart from the declarative user interface, `OFASys` provides an imperative user interface aiming for extensibility. Users can easily build experiments with their new ideas by extending "*base classes*". To integrate the custom components into the system, one needs to use the decorator `@register_config`. For detailed how-to tutorials, please refer to the `OFASys` documentation.

**Adding Task-Specific Data Processing Methods**   Although the modality presets can cover many of the data processing needs, one can further customize the data processing operations by extending the base class `BaseTask` with their own preprocessing/postprocessing logic. Those functions are called before/after the system preprocessor/postprocessor.

**Adding New Modality/Slot Type**   For adding new modality, one needs to implement a preprocessor and an adapter at the minimum. Preprocessors should extend the `SafeBasePreprocess` class, which contains a sanity check for the inputs. Adapters should extend the `BaseAdapter` class, which extends `torch.nn.Module`. The main difference between preprocessors/postprocessors and adapters from the library side is that adapters can be trained together with the model and their parameters must be saved in the model checkpoints.

## C   Detailed System Implementation

### C.1 Slot-wise Multi-Modal Data Processing for Composable Task Definition

As mentioned in Sec. 4.1, with existing frameworks, one has to organize the data for preprocessing differently w.r.t. each task, which is laborious and time-consuming. To address this issue, especially for multi-modal data, `OFASys` introduces slot-wise multi-modal processing realized by modality-specific preprocessors, postprocessors, and adapters. They

Table 3: Modality support in terms of slot types in `OFASys`.

| Slot Type | Adapter | Encoder | Decoder | Preprocessor | Postprocessor |
|---|---|:---:|:---:|---|---|
| TEXT | Embedding Lookup (Text) | ✓ | ✓ | text→token | text←token |
| | Embedding Lookup (Phone) | ✓ | ✗ | text→phone | N/A |
| IMAGE | Vision CNN/ViT | ✓ | ✗ | image→image3d | N/A |
| | Embedding Lookup (Image Code) | ✓ | ✓ | image→code | image←code |
| VIDEO | Vision CNN | ✓ | ✗ | image→video4d | N/A |
| AUDIO | Acoustic CNN | ✓ | ✓ | wave→fbank | wave←fbank |
| MOTION | Linear Projection | ✓ | ✓ | BVH→motion6d | BVH←motion6d, GIF←motion6d |
| BOX | Embedding Lookup (Box Code) | ✓ | ✓ | bounding box→code | bounding box←code |
| STRUCT | Embedding Lookup (Text) | ✓ | ✓ | schema→token, sudoku→token | sudoku←token |

conduct data transformation and feature extraction on a modality basis, which can be easily composed to define new multi-modal tasks. To be specific, `OFASys` includes a dispatcher that automatically sends the data in each slot to the corresponding preprocessor/postprocessor and adapter, according to the slot type, *i.e.*, its modality. The relation between slot types and slot-wise components is shown in Tab. 3.

Take Task 1 in Figure 2 as an example. The instruction is parsed as 5 slots: "[IMAGE]", "Which region does the text", "[TEXT]", "describes ?", and "[BOX]". The first slot is processed by the image-specific preprocessors and adapters. The three ones after essentially represent texts, and they are processed by the text-specific ones. The last one is processed by the ones for bounding boxes. By default, `OFASys` dispatches data to the processors of the corresponding modalities. This liberates users from the manual work on task-level modality-specific data processing.

**Preprocessor and Postprocessor** provide the ability to convert data from raw format to common machine learning data types and vice versa. Preprocessors take raw data as input, convert them into the proper form, and then collate multiple data example in a mini-batch into batched data, preparing for subsequent batch processing. Postprocessors convert model outputs to the original input format. In most cases, a postprocessor perform an inverse process of the corresponding preprocessor. For multiple slots of different modality in an instruction, `OFASys` includes a "*dispatcher*" to better manage the process. The dispatcher first applies the assigned preprocessor to the input data in each slot, and then the slots are processed in groups to improve efficiency. The slots from different examples are finally collated for batch processing in terms of their order in the instruction, *e.g.*, the data in the first image slot in the instruction from given examples are collated.

**Adapter** plays as the role of modality-specific feature extraction or representation learning. Adapters build a consistent input/output interface that unifies the difference among modalities for the universal model . Each "*input adapter*" takes the preprocessed data in the slot as input and outputs embedding/representation sequences. Input adapters can also produce auxiliary data needed by the model, *e.g.*, positional embeddings in terms of self-attention biases Raffel et al. (2020). If necessary, "*output adapters*" are implemented as an inverse process of the input adapters. Similar to preprocessors, `OFASys` includes a dispatcher to manage the process. The dispatcher applies the adapter to the batched preprocessed input and finally concatenates the representation sequences from slots to form the input of the universal model. As adapters contain trainable parameters, to avoid unnecessary memory costs in training, the adapters not used in the instructions are not initialized.

In existing application frameworks, the adapters in `OFASys` are often considered as part of the static overall model. However, adapters are by nature modality-specific and such practice limits the flexibility of multi-task learning on new modalities and new tasks. Therefore, `OFASys` disentangles modality-specific model computation from the universal backbone model computation. Yet a new problem arises that preprocessors/postprocessors and adapters are both modality-specific and one may implement data processing in either component. The difference in design is that adapters work on batched data and contain trainable parameters, which are saved in checkpoints.

### C.2 Modality-Agnostic Computation with Unified Representations

As the modality-specific data processing resolves the difference among modalities, the core model computation, namely the universal modal, can focus on the architecture design. `OFASys` can accommodate various universal model structures, including transformer-based sequence-to-sequence models Vaswani et al. (2017) like T5 Raffel et al. (2020), U-Net Ronneberger et al. (2015) for diffusion Ho et al. (2020), decoder-only models like GPT Brown et al. (2020); Chowdhery et al. (2022), *etc*. The only requirement for the universal model is that it shall take an embedding sequence as input and outputs another embedding sequence.

We currently provide a transformer-based encoder-decoder model as the default implementation of the universal model. Recent progress in different fields have witnessed the potential of the Transformer architecture becoming the universal framework, and sequence-to-sequence learning might be a paradigm towards generalist models Wang et al. (2022a); Reed et al. (2022b); Chowdhery et al. (2022); Wei et al. (2021); Alayrac et al. (2022b). Both the encoder and the decoder consist of Transformer blocks, each including self-attention, cross-attention, and point-wise feed-forward network (FFN). Additionally, inspired by Wang et al. (2022b; 2021), we implement a sparsely-activated Mixture-of-Experts (MoE) model as an alternative.

### C.3 Stage-wise Components

To build a complete computation task in training or inference, `OFASys` provides several stage-wise components, including criteria and generators. For each task, commonly-used criterions and generators provided by the system can be specified accordingly in the instruction. In the stage of training, the criterion computes the loss using the output of the above components and can be automatically deducted from the instruction. In the stage of inference, the generator is interpreted from the decoder slot in the instruction, which generates the final output with the help of output adapters and postprocessors.

**Criterion** forwards data through the model and uses the output of pre-/post-processors and adapters to calculate the loss for training. Several commonly used criteria are provided in `OFASys`. For example, CTC loss Graves et al. (2006) that is commonly considered in the ASR task for speech models Ao et al. (2022) can be declared on encoder slots. For outputs that are in the form of discrete tokens, *e.g.*, language tokens and image tokens van den Oord et al. (2017); Esser et al. (2021), by default, decoder slots perform softmax cross entropy within each associated vocabulary, similar to a language model. Besides token-level teacher forcing with the cross entropy criterion, the system provides other flexible learning paradigms, such as sequence-level loss, which supports reinforcement learning for action-reward tasks, and denoising diffusion probabilistic modeling (DDPM) Ho et al. (2020), one of the most commonly-used diffusion methods.

**Generator** is used to produce the final output using the model and provided data in inference. Generators in `OFASys` are separated into two categories according to the generation paradigm, *i.e.*, the "*auto-regressive generator*" and the "*diffusion generator*" Ho et al. (2020). The auto-regressive generator takes an auto-regressive approach to generation, which can be further divided into the discrete token generator and the continuous feature generator. The diffusion generator performs generation in a non-auto-regressive manner, where the model iteratively denoises the input to derive the output. The provided implementation covers all the supported modalities in generation.

### C.4 Flexible and Efficient Multi-Task Training with Schedulers

A critical challenge in multi-task learning is the scheduling of the tasks. In conventional practice, users need to first implement the aforementioned procedures of data and model processing task specifically, and then manually organize the training of multiple tasks with heuristic rules. For example, in implementation, Wang et al. (2022a) mixes the logic of all tasks in the code. If users want to add more tasks, they must be careful with the logic coupled with the previous tasks. The requirement in this scenario is the decoupling of task definitions and an efficient scheduler to train them together.

In `OFASys`, a task scheduler is responsible for training a model with multiple tasks. The execution of multi-task training is conceptually divided into two levels of abstractions in `OFASys`: the *logical scheduler* manages the strategy to compute the overall loss among tasks for each optimization step, while the *physical scheduler* determines how to partition and place the whole computation graph to physical devices with limited distributed capacity.

For the logical scheduler, users can implement their own strategies, such as those in multi-task learning Zhang & Yang (2021) or continual learning De Lange et al. (2021) literature, to decide either the task optimization order or the task importance. The default implementation provided by `OFASys` is a weighted average of all task losses, where users can adjust the task weight manually.

For the physical scheduler, efficient multi-modal multi-task training in distributed environments is highly challenging Barham et al. (2022). To illustrate, each task may activate a sub-part of the whole model, especially for the I/O adapters; the batch size and the sequence length of each task can also be different, resulting in uneven workloads between tasks. `OFASys` starts with a gradient accumulation-based scheduler. In one optimization step, each task performs forward and backward processes separately and accumulates gradients in the local device. Then, all devices reduce the gradients and update them for a step of optimization. Its advantage is that with the same configuration of each task, such as batch size and sequence length, the peak memory occupation of GPU devices can be the same as in single-task training. This implementation also supports each task using a different batch size.

## D  MODALITY SUPPORT

`OFASys` provides 7 presets for `TEXT`, `IMAGE`, `VIDEO`, `AUDIO`, `BOX`, `STRUCT`, and `MOTION` modality slots. Please note that the slots are categorized by its raw input data modality, not the transformed inner data modality.

### D.1  TEXT

`TEXT` is the most common slot type, as many other forms of structural data can be transformed into/from text, *e.g.*, category labels and table schemas. The default preprocessor conducts (sub-)tokenization using GPT2BPE Radford et al. (2019) and can apply masking for masked language modeling. There is another preprocessor that can transform text to phone used in audio tasks. The default postprocessor decodes tokens into text losslessly, as GPT2BPE is whitespace-aware. The adapter encodes and decodes the token or phone sequence using text embeddings or phone embeddings. As adapters of the same type share parameters, for sequence-to-sequence learning, it means source input embedding, target input embedding, and target output embedding are all shared. The available training objectives include MLE, SCST, and InfoNCE. As tokens are discrete, the currently supported generator is the auto-regressive token generator.

### D.2  IMAGE

`IMAGE` is another common slot type containing visual spatial data in different formats. The default preprocessor provides a simple series of transformations, such as resizing and normalization, to convert raw image data into the tensor format. We also provide a more complex preprocessor, which contains several data augmentation steps. `IMAGE` slot uses postprocessor only in image generation tasks and we use VQ-GAN Esser et al. (2021) to convert discrete code sequence back to images. We provide ResNet He et al. (2016) and ViT Dosovitskiy et al. (2021) as image adapters for encoder slots. For the decoder slot in image generation task, we use VQ-GAN to encode images into discrete code sequences. The available training objective is MLE using cross-entropy loss for image codes. Similar to `TEXT`, we use auto-regressive token generator to generate the discrete image code sequence.

### D.3  VIDEO

VIDEO is a slot type for consecutive image frames extracted from a video, *i.e.*, tempo-spatial data. The default preprocessor can decode the video and extract frames using a specified downsampling rate. It also supports data augmentations, following MViT Fan et al. (2021) and PySlowFast Fan et al. (2020). Currently, VIDEO can be only used as encoder slots. The adapter builds upon and reuses pretrained weights of the IMAGE adapter (ViT or ResNet) to encode each frame, and concatenate them to a sequence of representation vectors. The available training objectives include MLE and SCST. The VIDEO slot currently does not has a postprocessor or a generator. However, it is possible to cast the video generation task as image generation task: reuse the auto-regressive discrete generator to generate a single video frame and append the frame to the source; by repeating this step, we can generate a video.

### D.4  AUDIO

AUDIO is a slot type for sound features in the time and frequency domains. The default preprocessor can extract the log Mel-filterbank features from raw audio waveform. For robust speech preprocessing, techniques including (a) volume normalization and (b) cepstral mean and variance normalization (CMVN) Prasad & Umesh (2013) can be applied. Besides, specAugment Park et al. (2019) and speed perturbation Ko et al. (2017) are employed for data augmentation. AUDIO uses a postprocessor only in generation tasks, where we transform the predicted mel spectrograms to waveform via a vocoder, *i.e.*, HiFi-GAN Kong et al. (2020). Currently, the adapter can operate on log Mel-filterbank features. In the encoder slots, the AUDIO input adapter consists of a CNN for downsampling and a transformer network for contextutal representation learning. In the decoder slots, it employs a fully-connected network as the input adapter and a CNN as the output adapter following Tacotron2 Shen et al. (2018). The available training objectives include MSE of mel spectrograms and cross-entropy loss on the stop probability for speech synthesis. We utilize an auto-regressive feature generator for speech generation since fbank features are continuous.

### D.5  BOX

BOX is a slot type for processing bounding boxes in region-like tasks (*e.g.*, object detection, grounded captioning, and visual grounding). The default preprocessor can quantize the continuous corner coordinates (top-left and bottom-right) of the bounding box to discrete box tokens $\langle x_1, y_1, x_2, y_2 \rangle$ Chen et al. (2022a). The preprocessor allows the user to set the range of discrete values to control the granularity. BOX implements a postprocessor to recover the continuous corner coordinates from box tokens, so users can visualize the bounding boxes on the image. Since the coordinates have been converted to discrete box tokens, the TEXT adapter can be also used to encode the tokens into a sequence of embeddings. The available training objective is MLE. As box tokens are discrete, the currently-supported generator is the auto-regressive token generator.

### D.6  STRUCT

Structural data, such as databases, tables, grids, graphs, and trees, is widely used in many areas, *e.g.*, knowledge graph and protein structure. STRUCT is the slot for structural data, and currently, it supports table and database data. Inspired by UnifiedSKG Xie et al. (2022), the default preprocessor transforms the structural data into sequential text data. For tables of small sizes, the preprocessor flattens the whole table into a sequence, using ":" to distinct each column and "|" to distinct each row. For tables of large sizes or database schemas, the preprocessor only extracts the schemas information with a few mentioned row names from the instructions. Since STRUCT is only used as generation target in sudoku, we implement a text-to-sudoku postprocessor. Since the structural data are converted to text, the TEXT adapter can be used to encode the text tokens into a sequence of embeddings. The available training objective is MLE. As the transformed text tokens are discrete, the currently supported generator is the auto-regressive token generator.

## D.7 MOTION

Human motion, the MOTION modality, is common in 3D character animation, robotics, and human behavior understanding. We implement the "*motion_6d*" preprocessor for motion data. It reads a BVH file, which is a motion capture data format commonly used by the industry to describe a clip of motion, and converts the BVH file into a floating-point number (float) array of shape $[n, 6 + 6m]$, where $n$ is the number of frames and $m$ is the number of joints. Each frame describes the pose at that moment, which includes 3 floats describing the root joint's 3D position, another 3 floats for the body's velocity, and $6m$ floats corresponding to all the joints' rotations. Note that we convert each $3 \times 3$ rotation matrix into its 6D vector representation Zhou et al. (2019) to ease learning. The "*motion_6d*" adapter employs linear projection to align the dimension of the data and the universal model. By default, we use a criterion and a generator suitable for measuring the generation error of a continuous signal to handle motion data, i.e., the denoising diffusion probabilistic modeling (DDPM) Ho et al. (2020) loss and its corresponding generator. DDPM requires the adapter to handle the step information of DDPM, where the step information can also be understood as the strength of the noise for data corruption. The adapter incorporates this step information by adding a step embedding to the token embeddings, similar to how position embeddings are usually implemented.

## E  TASK EXAMPLES

OFASys currently includes 23 example tasks with default configurations. In the following, we summarize the default settings for those tasks.

### E.1  TEXT-ONLY TASKS

#### E.1.1  TEXT UNDERSTANDING (GLUE)

**Task Introduction:**  GLUE Wang et al. (2019) is a benchmark for text understanding, which casts multiple datasets into a unified sentence classification form. The tasks/datasets include the Corpus of Linguistic Acceptability (CoLA) Warstadt et al. (2019), the Stanford Sentiment Treebank (SST-2) Socher et al. (2013), Microsoft Research Paraphrase Corpus (MRPC) Dolan & Brockett (2005), Semantic Textual Similarity Benchmark (STS-B) Cer et al. (2017), Quora Question Pairs (QQP) Iyer et al. (2017), MultiNLI (MNLI) Williams et al. (2018); Bowman et al. (2015), Question NLI (QNLI) Rajpurkar et al. (2016); White et al. (2017); Demszky et al. (2018), Recognizing Textual Entailment (RTE) Dagan et al. (2005); Bar-Haim et al. (2006); Giampiccolo et al. (2007); Bentivogli et al. (2009), and Winograd NLI (WNLI) Levesque (2011). A majority of the original datasets are cast as natural lanugage inference (NLI) tasks (identifying entailment, neutral, contrast relationships) or binary classification tasks (yes/no). The evaluation metric is Matthew's Correlation Coefficient Matthews (1975) for CoLA, Pearson's and Spearman's Correlation Coefficient for STS-B, and accuracy for the rest. The overall score for this task in evaluation is commonly the arithmetic average over the 8 tasks without WNLI Devlin et al. (2019). Please note that currently OFASys is not able to support regression tasks in GLUE, *i.e.*, STS-B, which will be addressed soon, as there are no technical obstacles.

**Default Instruction:**  The default instructions for this task are as follows:

CoLA:

```
is the text "[TEXT:s]" grammatically correct?  -> is the text "[TEXT:s,no_loss]"
↪ grammatically correct? [TEXT:label,closed_set]
```

SST-2:

```
is the sentiment of text "[TEXT:s]" positive or negative? -> is the sentiment of text "[
↪ TEXT:s,no_loss]" positive or negative? [TEXT:label,closed_set]
```

QQP:

```
is question "[TEXT:q1]" and question "[TEXT:q2]" equivalent? -> is question "[TEXT:q1,
↪ no_loss]" and question "[TEXT:q2,no_loss]" equivalent? [TEXT:label,closed_set]
```

MRPC:

```
does text1 "[TEXT:s1]" and text2 "[TEXT:s2]" have the same semantics? -> does text1 "[
↪ TEXT:s1,no_loss]" and text2 "[TEXT:s2,no_loss]" have the same semantics? [TEXT:label,
↪ closed_set]
```

MNLI:

```
can text1 [TEXT:s1] imply text2 [TEXT:s2]? -> can text1 [TEXT:s1,no_loss] imply text2 [
↪ TEXT:s2,no_loss]? [TEXT:label,closed_set]
```

QNLI:

```
does "[TEXT:s]" contain the answer to question "[TEXT:q]"? -> does "[TEXT:s,no_loss]"
↪ contain the answer to question "[TEXT:q,no_loss]"? [TEXT:label,closed_set]
```

RTE:

```
can text1 "[TEXT:s1]" imply text2 "[TEXT:s2]"? -> can text1 "[TEXT:s1,no_loss]" imply
↪ text2 "[TEXT:s2,no_loss]"? [TEXT:label,closed_set]
```

The input sentence(s) are repeated for better performance, as observed in Wang et al. (2022a). `no_loss` indicates the repeated input sentences do not have targets in computing the loss. The generator in inference is the sequence generator with the token space constrained to a dataset-specific space via `closed_set`.

### E.1.2  TEXT SUMMARIZATION (GIGAWORD)

**Task Introduction:**  Text summarization is a natural language generation task, where the model should produce a piece of concise text that covers the main points of the given text. `OFASys` currently evaluates text summarization on Gigaword, following related work Rush et al. (2015). Gigaword for summarization is a naturally-annotated dataset consisting of news articles, where the first sentence of the article is regarded as the summary for the rest of the first paragraph. The evaluation metric is ROUGE Lin (2004). Especially, we report ROUGE-L (R-L) in this paper.

**Default Instruction:**  The default instruction for this task is as follows:

```
what is the summary of article " [TEXT:src] "? -> [TEXT:tgt,noise_ratio=0.2]
```

The `noise_ratio` is passed to the preprocessor, which randomly replaces tokens with the specified rate in the target output.

### E.1.3  NATURAL-INSTRUCTIONS V2

**Task Introduction:**  Natural-Instructions v2 Wang et al. (2022c) is a benchmark of over 1600 diverse language tasks which evaluates generalization across language tasks by leveraging their language instructions. It covers over 70 distinct task types, such as tagging, in-filling and rewriting. These tasks are collected with contributions of NLP practitioners in the community and through an iterative peer review process to ensure their quality. Natural-Instructions v2 consists of a variety of language tasks and instructions that describe them in plain language. Each sample contains four fields. Instruction defines a given task in plain language. This involves a complete definition of how an input text (*e.g.*, a sentence or a document) is expected to be mapped to an output text. Examples are samples of inputs and correct or wrong outputs to them, along with a short explanation for each. On average, each sample contains 2.8 positive and 2.4 negative examples. Src and tgt are a large collection of input-output pairs for each task. Since this benchmark contains a large collection of tasks, we split the tasks into two subsets: one subset for evaluation and the remaining ones which

can be used for supervision. For evaluation tasks, specifically, we fix a manually-selected collection of 12 categories that represent 154 tasks. We report ROUGE-L Lin (2004) for aggregated performance results across a variety of tasks which is a soft string overlap metric that can be applied to a wide range of text generation tasks.

**Default Instruction:** The default instruction for this task is as follows:

```
[TEXT:instruction] [TEXT:examples] [TEXT:src] -> [TEXT:tgt,max_length=128]
```

The maximum input length is set to 1024 and the output `max_length` is set to 128.

### E.1.4 TEXT INFILLING

**Task introduction:** Self-supervised learning in natural language processing can belong to various forms. In `OFASys`, we give an example for the most basic form of self-supervised learning for sequence-to-sequence models, that is, text-infilling, in a similar manner to BART Lewis et al. (2020). For this task, the tokens in the input text is randomly replaced with a special mask token, and the model should recover the input text as the output. In our experiments, the data is obtained from English Wikipedia. This task is designed for pretraining only.

**Default instruction:** The default instruction for this task is as follows:

```
what is the complete text of "[TEXT:text,mask_ratio=0.3]"? -> [TEXT:text]
```

The `mask_ratio` is passed to the preprocessor, which randomly replaces input tokens with a special mask token.

## E.2 IMAGE-RELATED TASKS

### E.2.1 IMAGE CLASSIFICATION (ILSVRC-2012)

**Task Introduction:** Image classification task requires the model to predict the correct category for the input image. We evaluate our model on the ILSVRC-2012 ImageNet dataset Russakovsky et al. (2015). The dataset contains 1 k image categories and around 1.3 M images. Each image is manually annotated with one category label among the 1 k candidates. We report the top-1 accuracy on the test set of 50 k images.

**Default Instruction:** The default instruction for this task is as follows:

```
[IMAGE:image,preprocess=imagenet] what does the image describe? -> [TEXT:label_name,
↪ closed_set]
```

The input image resolution is set to 480. For the input image, we specify a special ImageNet preprocessor to activate the dataset-specific image augmentation on training samples. Specifically, following Bao et al. (2022), we employ the same random resize cropping, random flipping, RandAug Cubuk et al. (2020) and random erasing Zhong et al. (2020b) as data augmentation strategies on the training images. For the decoder slot, we add the specification `closed_set` to constrain the text output into the 1K candidate category names.

### E.2.2 IMAGE CAPTIONING (COCO CAPTIONS)

**Task Introduction:** Image captioning requires the model to generate a descriptive text for an image. We evaluate the multi-modal generation capability of `OFASys` on the most widely used COCO Caption dataset Chen et al. (2015). Following previous works Anderson et al. (2018); Wang et al. (2022a), We report CIDEr Vedantam et al. (2015) scores on the Karpathy test split Karpathy & Fei-Fei (2015).

**Default Instruction:** The default instruction for this task is as follows:

```
[IMAGE:img] what does the image describe? -> [TEXT:cap]
```

### E.2.3  VISUAL ENTAILMENT (SNLI-VE)

**Task Introduction:**  Visual entailment (VE) Xie et al. (2019) is similar to textual entailment. It changes the premise from the text to the image, and judges whether the images matches the sentence. SNLI-VE is a data set of VE tasks which gives images, image captions and premises, and requires the model to judge the relationship between images and premises, and gives one of three outcomes: entailment, neutral, and contradiction.

**Default Instruction:**  The default instruction for this task is as follows:

```
[IMAGE:img] can image and text1 "[TEXT:cap]" imply text2 "[TEXT:hyp]"? -> can image and
↪ text1 "[TEXT:cap,no_loss]" imply text2 "[TEXT:hyp,no_loss]"? [TEXT:label,closed_set]
```

The pattern is similar to MNLI task, where the encoder text is repeated in the decoder. The input image resolution is resized to $480 \times 480$ by default.

### E.2.4  VISUAL QUESTION ANSWERING (VQA V2)

**Task Introduction:**  Visual question answering (VQA) requires the model to answer questions based on the information of the given image Antol et al. (2015); Goyal et al. (2017). We finetune our pretrained model on the dataset VQA v2 Goyal et al. (2017). We evaluate the performance by calculating accuracy.

**Default Instruction:**  The default instruction for this task is as follows:

```
[IMAGE:image] [TEXT:question] -> [TEXT:answer,closed_set]
```

The resolution of the input images is $480 \times 480$. The generated string is constrained to a closed_set, similar to the formulation of classification tasks.

### E.2.5  VISUAL GROUNDING (REFCOCO)

**Task Introduction:**  Visual grounding requires the model to locate an image region based on a textual description. OFASys formulates this task as a sequence-to-sequence generation task. The model takes a text and an image as input and generates a sequence of box tokens in an autoregressive manner. We perform experiments on RefCOCO Yu et al. (2016); Kazemzadeh et al. (2014). The standard metric Acc@0.5 is reported on the corresponding validation set, that is, a bounding box is considered correct if IoU $\geq 0.5$ with the ground truth.

**Default Instruction:**  The default instruction for this task is as follows:

```
[IMAGE:img] which region does the text "[TEXT:cap]" describe? -> [BOX:region_coord]
```

### E.2.6  GROUNDED IMAGE CAPTIONING

**Task Introduction:**  Grounded image captioning is the inverse task of visual grounding. Given an image and a region, the model is required to generate a description about the region. We use RefCOCO, RefCOCO+, RefCOCOg, and Visual Genome Krishna et al. (2017) as the pretraining datasets for this task. This task is supposed to used in multi-task learning only and no inference support is currently included.

**Default Instruction:**  The default instruction for this task is as follows:

```
[IMAGE:img] what does the region describe? region [BOX:region_coord] -> [TEXT:cap]
```

### E.2.7 Object Detection

**Task Introduction:** Object detection is a common vision task that requires a model to recognize all objects in the image and localize their regions. We use OpenImages Kuznetsova et al. (2020), Object365 Shao et al. (2019), Visual Genome Krishna et al. (2017), and COCO Chen et al. (2015) as the pretraining datasets for this task. This task is supposed to used in multi-task learning only and no inference support is currently included.

**Default Instruction:** The default instruction for this task is as follows:

```
[IMAGE:img] what are the objects in the image? -> [ [BOX] [TEXT] ]*
```

As the output is of variable lengths, a specific instruction parsing method is implemented via the `build_instruction` method. The "[]*" notation is parsed to the actual number of the bounding boxes for a training example.

### E.2.8 Image Infilling

**Task Introduction:** Image infilling task has been proved an effective task for both image and multi-modal pretraining He et al. (2022); Bao et al. (2022); Wang et al. (2022a). We mask the middle part of the raw images as input, and expect the model to restore the masked part from the corrupted input by generating the discrete codes produced by VQ-GAN models. This self-supervised task is designed for multi-task pretraining only.

**Default Instruction:** The default instruction for this task is as follows:

```
what is the complete image of "[IMAGE:img,mask_ratio=0.5]"? -> [IMAGE,
↪ preprocessor=image_vqgan,adapter=image_vqgan]
```

The input is the representation from the image pixels. The input image resolution is set to $256 \times 256$ and we mask the central $128 \times 128$ part. The attribute `mask_ratio` is added to set the mask ratio of the image. Following Wang et al. (2022a), the target output is a seqeunce of discrete image codes generated by VQ-GAN Esser et al. (2021). The output length in inference is fixed to 256 ($16 \times 16$), according to the image resolution and the compression ratio of VQGAN.

### E.2.9 Image Generation (COCO Captions)

**Task Introduction:** Text-to-Image generation has become a task that has attracted more and more attention of researchers as it demonstrates the excellent creation of neural network models Ramesh et al. (2021); Ding et al. (2021); Rombach et al. (2021). Similar to Image Infilling task, we use a VQ-GAN model to convert images into discrete codes, so that the sequence generator can generate a complete image by generating the code sequence autoregressively. We train our model on the train split of the MS COCO dataset and evaluate our model on the test split by randomly sampling 30 000 images. As for evaluation, following previous works Wang et al. (2022a); Wu et al. (2022); Huang et al. (2021), we use CLIP Similarity Score (CLIPSIM) to evaluate the semantic similarity between the query text and the generated images.

**Default Instruction:** The default instruction for this task is as follows:

```
what is the complete image? caption: "[TEXT:cap]"? -> [IMAGE,preprocessor=image_vqgan,
↪ adapter=image_vqgan]
```

We use the similar instruction and configuration like image infilling task to define the image generation task. The main difference is that we use text instead of masked images as the input. In inference, the output image resolution is set to $256 \times 256$, so the output length is 1024 ($32 \times 32$) with respect to the compression ratio of VQ-GAN.

### E.3 Video-Related Tasks

#### E.3.1 Video Classification (Kinetics-400)

**Task Introduction:** The video classification task is a fundamental task in the field of video understanding where the model needs to predict the label for a given video clip. We evaluate our model on the Kinetics-400 dataset Kay et al. (2017), which contains 300 k video clips from 400 classes. We report the accuracy on the val split of the Kinetics-400 dataset.

**Default Instruction:** The default instruction for this task is as follows:

```
[VIDEO:video] what does the video describe? -> [TEXT:label_name,closed_set]
```

We follow MViT Fan et al. (2021) for video data augmentation, which is incorporated into the `VIDEO` preprocessor.

#### E.3.2 Video Captioning (MSR-VTT)

**Task Introduction:** The video captioning task requires the model to generate a textual description for a given video clip. We evaluate the proposed method on MSR-VTT caption dataset Xu et al. (2016), which contains 10 k video clips 200 k descriptions of the videos. Following Lin et al. (2022), we report CIDEr Vedantam et al. (2015) scores on the val split of the MSR-VTT dataset.

**Default Instruction:** The default instruction for this task is as follows:

```
[VIDEO:video] what does the video describe? -> [TEXT:cap]
```

All data augmentation for videos are disabled except for the random flip.

#### E.3.3 Video Question Answering (MSR-VTT QA)

**Task Introduction:** The video captioning task requires the model to generate a answer for a given video clip and a question related to that video clip. We evaluate the proposed method on MSR-VTT QA dataset Xu et al. (2017), which contains question-answer pairs extracted from the original MSR-VTT dataset Xu et al. (2016). We report the accuracy on the val split of MSR-VTT QA dataset.

**Default Instruction:** The default instruction for this task is as follows:

```
[VIDEO:video] [TEXT:question] -> [TEXT:answer,is_label]
```

We follow MViT Fan et al. (2021) for video data augmentation.

### E.4 Audio-Related Tasks

#### E.4.1 Automatic Speech Recognition

**Task Introduction:** Automatic Speech Recognition (ASR) is the task of converting speech into sequences of discrete semantic tokens. We evaluate our model on the Librispeech Panayotov et al. (2015) and AISHELL-1 Bu et al. (2017) dataset. The Librispeech dataset contains 1000 hours of speech in English sampled at 16 kHz. The AISHELL-1 dataset contains 178 hours of Mandarin speech sampled at 16 kHz.

**Default Instruction:** The default instruction for this task is as follows:

```
[AUDIO:wav] what is the text corresponding to the voice? -> [TEXT:text]
```

To achieve better performance, the criterion for this tasks include an additional Connectionist Temporal Classification (CTC) Watanabe et al. (2017) loss alongside the sequence-to-sequence

loss. Specifically, we input the encoder output matrix $X$ and the target sequence $Y$ for CTC loss, which computes $P(Y|X)$ by summing over the probability of all possible alignments between the two and maximizes the probability of $P(Y|X)$.

### E.4.2 TEXT-TO-SPEECH

**Task Introduction:** Text-to-speech (TTS) is the task of generating speech from input text. We evaluate our model on the LJSpeech Ito & Johnson (2017) and BZNSYP[3] datasets. The LJSpeech dataset contains 24 hours of English audio of a single speaker reading passages with a sample rate of 22 050 Hz. The BZNSYP dataset includes 12 hours of Mandarin audio sampled at 48 kHz from a single speaker.

**Default Instruction:** The default instruction for this task is as follows:

```
[TEXT:text,preprocessor=text_to_phone] what is the voice corresponding to the text? -> [
↪ AUDIO:fbank,adapter=audio_tgt_fbank]
```

`OFASys` converts original text into phonemes, then take phonemes as model input and output mel spectrograms. In inference, we use a well-trained vocoder HiFi-GAN Kong et al. (2020) to transform the predicted mel spectrograms to the waveform.

### E.5 STRUCTURAL DATA–RELATED TASKS

### E.5.1 TEXT-TO-SQL (SPIDER)

**Task Introduction:** Text-to-SQL could be considered as a semantic parsing task, which aims to generate executable SQL codes according to the question text and the information of corresponding database. In this task, model is supposed to not only truly understand the question text and database but also generate a SQL format code to solve the question. We conduct our experiments on Spider dataset Yu et al. (2018), which contain different complex SQL queries and different complex database in different domains. It consists of 10 181 questions and 5693 unique complex SQL queries on 200 databases with multiple tables, covering 138 different domains. Some questions in the dataset is supposed to answer by cross-domain and cross-database semantic parsing problems. We use Exact Matching metric, measuring whether the generated SQL code as a whole is equivalent to the label SQL query. Following previous Text-to-SQL studies Zhong et al. (2020a), we first decompose the SQL of both prediction and ground truth as bags of several components (SELECT, WHERE, GROUP BY, ORDER BY, KEYWORDS) and sub-components. The generated SQL code is correct only if all the components are correct compared with ground truth. The Exact Matching metric is the ratio of correct prediction among all the predictions.

**Default Instruction:** The default instruction for this task is as follows:

```
[TEXT:src]; structured knowledge: "[STRUCT:database,preprocessor=database_to_text]".
↪ generating sql code. -> [TEXT:tgt]
```

Similar with Xie et al. (2022), we consider the task as a sequence to sequence language task. `src` slot is the question text, and the `database` slot is the corresponding text format database information of the samples. The database information is transformed by `table_to_text` into

```
| [database name] | [table_1 name] : [column_1 name] ( [mentioned row names] ), [column_2
 name], [column_3 name], ... [table_2 name] : ...
```

The `mentioned row names` are the rows which are mentioned in the question. Adding the mentioned row names improves the performance in most cases Xie et al. (2022).

### E.5.2 TABLE-TO-TEXT (DART)

**Task Introduction:** Table-to-Text Cawsey et al. (1997); Lebret et al. (2016) aims to describe a table by natural language. We conduct experiments on DART Nan et al. (2021),

---

[3]https://www.data-baker.com/open_source.html

which is a triplet component table dataset. We consider the triplets as a three-column, multi-row table without column names. DART has 62 659, 5980, and 12 552 examples for training, validation, and testing, respectively. The evaluation metric of Table-to-Text is BLEU Papineni et al. (2002) from SacreBLEU Post (2018).

**Default Instruction:**    The default instruction for this task is as follows:

```
structured knowledge: "[SRTUCT:database,preprocessor=table_to_text]". how to describe the
↪   tripleset? -> [TEXT:tgt]
```

We consider the task as a sequence-to-sequence language task, where database is the table information following the format as

```
[row1 col1] : [row1 col2] : [row1 col3] | [row2 col1] : [row2 col2] : [row2 col3] | ...
```

### E.5.3   TABLEQA (FETAQA)

**Task Introduction:**    TableQA Jin et al. (2022) is a question answering task according to a given table. We use FeTaQA dataset Nan et al. (2022) to evaluate our methods. FeTaQA is dataset based on 10 k Wikipedia entries of (table, question, free-form answer, supporting table cells). We only use the table, question and free-form answers. The evaluation of TableQA is BLEU Papineni et al. (2002).

**Default Instruction:**    The default instruction for this task is as follows:

```
structured knowledge: "[STRUCT:database,preprocessor=table_to_text]". what is the answer
↪ of the question "[TEXT:src]"? ->  [TEXT:tgt]
```

The `src` is the question, the `database` is the table, and the `tgt` is the predict answer. The table format is the same as Table-to-Text.

### E.5.4   SUDOKU

**Task Introduction:**    Sudoku is a common math puzzle game, which fills the blank of a $9 \times 9$ tables with digits 1-9, such that each digit appears exactly once in each row, column, and $3 \times 3$ box. Normally, a Sudoku has a single unique solution. We use the Sudoku dataset in Kaggle[4], which contains 10 M puzzles, with difficulty from easy to hard. The dataset is randomly split into 1000 samples for validation and 1000 for testing with the rest used for training. We use Solved Acc as the evaluation metrics, which means the prediction that meets all the requirements is considered correct.

**Default Instruction:**    The default instruction for this task is as follows:

```
"[STRUCT:src,preprocessor=sudoku_to_text]". solve the sudoku. -> [STRUCT:tgt,
↪ preprocessor=sudoku_to_text]
```

The `src` is the sudoku puzzles. `sudoku_to_text` uses ":" to split columns and "|" to split rows into the form like

```
0 : 8 : 5 : 2 : 3 : 0 : 0 : 7 : 0 |
1 : 4 : 0 : 8 : 0 : 9 : 0 : 0 : 0 |
0 : 7 : 0 : 0 : 1 : 0 : 0 : 0 : 8 |
7 : 0 : 9 : 0 : 0 : 5 : 0 : 0 : 3 |
0 : 0 : 0 : 1 : 6 : 0 : 0 : 0 : 0 |
5 : 0 : 2 : 3 : 0 : 0 : 0 : 1 : 0 |
0 : 0 : 1 : 7 : 4 : 8 : 0 : 5 : 9 |
6 : 5 : 0 : 9 : 0 : 3 : 0 : 0 : 0 |
8 : 9 : 0 : 6 : 0 : 0 : 7 : 0 : 2 |
```

In the sequence, 0 means blank. The `tgt` is the same format as `src`, replacing the 0 with answers.

---

[4]https://www.kaggle.com/datasets/rohanrao/sudoku

Table 4: Tasks, datasets, and main optimization hyper-parameters used in the experiments. We list the configurations of single-task and multi-task settings.

| Task | Dataset | Specialist | | Generalist & Generalist MoE |
| --- | --- | --- | --- | --- |
| | | Batch Size | LR | Batch Size |
| *Text only tasks* | | | | |
| Instruction Tuning | NaturalInstruction v2 | 512 | $1 \times 10^{-5}$ | 512 |
| Summarization | Gigaword | 512 | $1 \times 10^{-4}$ | 512 |
| *Image tasks* | | | | |
| Classification | ILSVRC | 256 | $5 \times 10^{-5}$ | 2048 |
| Visual Entailment | SNLI-VE | 256 | $2 \times 10^{-5}$ | 256 |
| Captioning | COCO | 128 | $1 \times 10^{-5}$ | 2048 |
| Visual Grounding | RefCOCO | 128 | $3 \times 10^{-5}$ | 2048 |
| Grounded Caption | RefCOCO | 256 | $1 \times 10^{-5}$ | 256 |
| VQA | VQA v2 | 512 | $5 \times 10^{-5}$ | 1536 |
| Image Generation | COCO | 512 | $1 \times 10^{-3}$ | 512 |
| *Audio tasks* | | | | |
| ASR | LibriSpeech | 256 | $1 \times 10^{-3}$ | 2048 |
| TTS | LJSpeech | 256 | $1 \times 10^{-3}$ | 1024 |
| *Video tasks* | | | | |
| Classification | Kinetics400 | 512 | $5 \times 10^{-5}$ | 512 |
| Captioning | MSR-VTT | 256 | $5 \times 10^{-5}$ | 128 |
| VQA | MSR-VTT QA | 512 | $5 \times 10^{-5}$ | 256 |
| *Motion tasks* | | | | |
| Text-to-Motion | AMASS/KIT/AIST++ | 512 | $1 \times 10^{-3}$ | 2048 |
| *Other tasks* | | | | |
| Table-to-Text | DART | 192 | $5 \times 10^{-5}$ | 128 |
| Text-to-SQL | Spider | 64 | $5 \times 10^{-5}$ | 256 |

## E.6 MOTION-RELATED TASK

### E.6.1 TEXT-TO-MOTION SYNTHESIS

**Task Introduction:** The text-to-motion synthesis task requires the model to generate a clip of human motion meeting the description of the given text. We use the text-motion pairs provided by KIT Plappert et al. (2016). The AMASS Mahmood et al. (2019) dataset is also used to enrich the training data. The text is set to an empty string when using AMASS, since AMASS does not provide text labels. Unlike previous works Lin et al. (2018); Ahuja & Morency (2019), we learn the rotation parameters rather than the positions of all the joints, in order to produce visually better results, which unfortunately means that we cannot directly compare our approach to the previous methods on the same benchmark.

**Default Instruction:** The default instruction for the text-to-motion task is as follows:

```
motion capture: [TEXT:title] -> [MOTION:bvh_frames]
```

The training criterion and the inference generator follow the DDPM method. The accompanied preprocessor, adapter, and postprocessor are also available. Please refer to Appendix D.7 for more details.

## F MORE EXPERIMENTAL SETTINGS

The task mixture used in multi-task training include:

1. Text summarization on Gigaword Rush et al. (2015)
2. Instruction tuning on Natural-Instructions v2 Wang et al. (2022c)

3. Image classification on ImageNet-1K Russakovsky et al. (2015)
4. Image captioning on COCO Caption Chen et al. (2015)
5. Visual entailment on SNLI-VE Xie et al. (2019)
6. Visual grounding on RefCOCO Yu et al. (2016)
7. Grounded image captioning on RefCOCO Yu et al. (2016)
8. Image generation on COCO Caption Chen et al. (2015)
9. Visual question answering on VQA v2 Goyal et al. (2017)
10. Video classification on Kinetics400 Russakovsky et al. (2015)
11. Video captioning on MSR-VTT Xu et al. (2016)
12. Video question answering on MSR-VTT QA Xu et al. (2017)
13. Automatic speech recognition on LibriSpeech Panayotov et al. (2015)
14. Text-to-speech on LJSpeech Ito & Johnson (2017)
15. Table-to-text on DART Nan et al. (2021)
16. Text-to-SQL on Spider Yu et al. (2018)
17. Text-to-motion synthesis on AMASS Mahmood et al. (2019), KIT Plappert et al. (2016) and AIST++ Li et al. (2021)

2 of these tasks, *i.e.*, grounded image captioning and text-to-motion synthesis, does not come with validation sets and as a result, no scores are reported.

The optimization settings for those tasks are listed in Tab. 4. For multi-task learning, as shown in Sec. 5, a learning rate of $3 \times 10^{-4}$ is used. The batch size is altered to better suit the multi-task learning settings. For single-task learning, each specialist is trained with a more appropriate set of hyper-parameters.

