# OpenReview forum: "OFASys: A Multi-Modal Multi-Task Learning System for Building Generalist Models"
_ICLR.cc/2024/Conference — ICLR 2024 Conference Withdrawn Submission_

### Official Review · Reviewer_G7DQ · 2023-10-30

**Soundness:** 3 good
**Presentation:** 3 good
**Contribution:** 3 good
**Rating:** 1
**Confidence:** 5

**Summary:**

While I appreciate the effort and work that has gone into your submission, I would like to bring to your attention a significant oversight regarding the submission guidelines.

ICLR has a [strict policy](https://iclr.cc/Conferences/2024/CallForPapers) that mandates a maximum of 9 pages for the main text of the submission, excluding citations and appendices. Upon reviewing your paper, I noticed that the main text exceeds this limit, spanning 10 pages.

Adherence to conference guidelines is crucial not only for maintaining a consistent standard across all submissions but also for ensuring fairness in the review process. Overstepping the page limit can be perceived as a lack of attention to detail, which might impact the overall evaluation of the paper.

I strongly recommend revisiting the conference guidelines before your submisson.

**Strengths:**

refer to the summary section

**Weaknesses:**

refer to the summary section

**Questions:**

refer to the summary section

---

### Official Review · Reviewer_4hDS · 2023-10-31

**Soundness:** 2 fair
**Presentation:** 3 good
**Contribution:** 2 fair
**Rating:** 5
**Confidence:** 4

**Summary:**

The article introduces OFASys, a multi-modal generalist model learning system. The core concept of OFASys is the decoupling of multi-modal task representations from underlying model implementations. It leverages a declarative task interface called "multi-modal instruction", allowing a task involving multiple modalities to be defined with just a single line of code. The system automatically generates task plans for both training and inference, and supports multi-task training for diverse multi-modal workloads.

**Strengths:**

1. The article is relatively clear and easy to understand.
2. It integrates multiple modalities, including both multi-modal and uni-modal, into a single framework and achieves commendable performance.

**Weaknesses:**

1. The core question raised is why there's a need to forcibly merge different tasks from multiple modalities into a single framework. The current method integrates various modalities and tasks using MOE (Mixture of Experts), but it seems to only reduce some parameters without bringing clear benefits; instead, it may lead to performance loss. It's suggested that the authors should consider directions where different tasks compete and promote each other.
2. The abstract mentions, “OFA+ model achieves 95% performance in average with only 16% parameters of 15 task-finetuned models.” It's queried whether this 16% refers to the sum of parameters from all 15 tasks. If so, many of the 15 tasks listed in the article could be merged, and better performance could be achieved through a lightweight Adapter or LoRA, for instance, merging Image Classification with Video Classification.
3. Additionally, the unified framework appears to be achieved through extensive engineering design rather than a unification like Large Language Models (LLMs) through instructions. There are already many works related to multi-modal LLMs (e.g., PandaGPT) that might be closer to the concept of "multi-modal instruction."

**Questions:**

Please see weaknesses.

---

### Official Review · Reviewer_DATQ · 2023-11-02

**Soundness:** 3 good
**Presentation:** 3 good
**Contribution:** 2 fair
**Rating:** 3
**Confidence:** 3

**Summary:**

This paper proposes a generalist model learning system which can combine several modalities and tasks together into just one system. It seems quite convenient to define a task involving multiple modalities using just one line of instruction. Meanwhile, two versions of generalist models based on this system are trained.

**Strengths:**

1. Experimental results are presented on popular datasets and common tasks, such as summarization, visual grounding, TTS in Table 2.
2. This is a work that requires substantial engineering efforts. This open-source system may facilitate multi-modal and multi-task training in the community.

Overall I think the paper is more like an engineering practice. There is in lack of a specific explanation for its novelty. Many details on codes and how to use them make the paper quite confusing and a little bit hard to follow.

**Weaknesses:**

1. On page 4 and page 5, Focusing on the details of how to use this system, rather than the intrinsic insights, confuses readers and makes this paper difficult to understand.

2. The details on how to use data from multiple downstream tasks for training have not been described, in 5.1. Even in Appendix F, I didn't find how you combine the data of different tasks, by which proportion, and using what kind of combination. It's not clear what advantage this could bring about.

**Questions:**

What's the difference between your model OFA+ and the Gato of Deepmind [1]? The two models are both designed to handle multi-modalities and multi-tasks, convert the multi-modal inputs into tokens, and take the use of transformer decoder structure as the universal model, as you depict the setting of your model in 5.1.

[1] Reed, Scott, et al. "A generalist agent." arXiv preprint arXiv:2205.06175 (2022).